# Dereplication of Components Coupled with HPLC-qTOF-MS in the Active Fraction of *Humulus japonicus* and It’s Protective Effects against Parkinson’s Disease Mouse Model

**DOI:** 10.3390/molecules24071435

**Published:** 2019-04-11

**Authors:** Hee Ju Lee, Basanta Dhodary, Ju Yong Lee, Jin-Pyo An, Young-Kyoung Ryu, Kyoung-Shim Kim, Chul-Ho Lee, Won Keun Oh

**Affiliations:** 1Korea Bioactive Natural Material Bank, Research Institute of Pharmaceutical Sciences, College of Pharmacy, Seoul National University, Seoul 08826, Korea; hjlee81@kist.re.kr (H.J.L.); Basanta02@hotmail.com (B.D.); sbplee@snu.ac.kr (J.Y.L.); ntopjp77@gmail.com (J.-P.A.); 2Natural Product Informatics Research Center, Korea Institute of Science and Technology, Gangneung 25451, Korea; 3Laboratory Animal Resource Center, Korea Research Institute of Bioscience and Biotechnology (KRIBB), Daejeon 34141, Korea; mnc1208@kribb.re.kr (Y.-K.R.); kskim@kribb.re.kr (K.-S.K.); chullee@kribb.re.kr (C.-H.L.)

**Keywords:** *Humulus japonicus*, HPLC-qTOF-MS, *in vitro* MAO B assay, in vivo 6-OHDA-lesioned PD animal test, Parkinson’s Disease

## Abstract

*Humulus japonicus* is an annual plant belonging to the Cannabacea family, and it has been traditionally used to treat pulmonary tuberculosis, dysentery, chronic colitis, and hypertension. We investigated the active components against Parkinson’s disease from *H. japonicus* fraction (HJF) using high performance liquid chromatography (HPLC) coupled with quadruple-time-of-flight mass spectroscopy (qTOF-MS) and NMR. Fourteen compounds were isolated from HJF, including one new compound, using HPLC-qTOF-MS and NMR. The major compounds of HJF were luteolin-7-*O*-glucoside and apigenin-7-*O*-glucoside, and there was approximately 12.57- and 9.68-folds increase in the contents of these flavonoids compared to those of the 70% EtOH extract. Apigenin and luteolin exhibited the strongest inhibitory effects on monoamine oxidase (MAO) B enzyme activity. In animal studies, limb-use behavior was significantly reduced by unilateral 6-OHDA lesion and ipsilateral rotations. These results indicated that oral administration of 300 mg/kg HJF resulted in the improvement of motor asymmetry and motor impairment in unilateral 6-OHDA-lesioned mice. HJF, including active components leads to an improvement of motor behavior in a Parkinson’s disease mouse model.

## 1. Introduction

Parkinson’s disease (PD) is a common neurodegenerative disorder that caused motor problems such as resting tremor, muscle rigidity, and postural instability [1]. PD pathogenesis is associated with genetic disposition, age and environmental factors, mitochondrial dysfunction, oxidative damage, inflammation, and microgliosis [2,3]. Oxidative damage by the generation of reactive oxygen species (ROS) causes neuronal membrane damage, including damage to membrane proteins, unsaturated lipids, and DNA [4]. Recent studies have suggested that the side chains of membrane proteins and lipids are modified by ROS, and a reduction in membrane unsaturation is associated with decreased membrane fluidity, enzyme concentration, and receptor concentrations [5]. The brain is susceptible to oxidative stress damage due to the high concentration of polyunsaturated fatty acids and oxygen. Oxidative damage by 6-hydroxydopamine (6-OHDA) induces apoptosis in various neurons [6], and toxicity and apoptosis by 6-OHDA are induced by activation of the mitogen-activated protein kinase (MAPK) pathway, which includes p38 kinase, extracellular signal-regulated kinase 1/2 (ERK 1/2), and c-Jun N-terminal kinase (JNK) [7,8].

Monoamine oxidase (MAO), a FAD-dependent enzyme, is a catalyst for the oxidative deamination of several amines and is present in various mammalian organs with two distinct subtypes, MAO A and MAO B. MAO removes the NH_2_ group of monoamine to produce CHO group and hydrogen peroxide [9]. The level of MAO B increases in the brain of aging or PD patients [10], and dopamine (DA) is mainly oxidized by human MAO B [11]. Previous studies have indicated that increased DA oxidation in the elderly is associated with dopaminergic cell death in the substantia nigra, a hallmark of PD [12]. MAO B inhibitors have been shown to delay the progress of symptoms and to have neuroprotective effects from programed cell death due to ROS [13]. MAO B inhibition also prevents nigrostriatal neuron cell death due to oxidative damage. Thus, MAO B inhibitors have been used as medications to protect neurons from oxidative agents and neurodegenerative diseases [14,15].

*Humulus japonicus* Sieboid & Zucc (Cannabaceae) (www.theplantlist.org) (synonym *Humulus scandens*) is a vine plant that grows in many regions of Asia, including China, Japan, and Korea. This plant is an ornamental plant known as a wild hop or a Japanese hop and is managed as a harmful plant in many countries because it grows very quickly in new habitats and hinders the growth of other plants. However, this plant has been used in traditional Korean medicine for the treatment of pulmonary tuberculosis, dysentery, chronic colitis, and hypertension. The constituents from this plant have also been reported to have antioxidant and antibacterial activities as well as anti-tumor effects [16]. 

Recent study has shown that the 70% EtOH extract of *H. japonicus* protects against oxidative stress *in vivo* and *in vitro* models of Parkinson’s disease [17] and attenuates neuroinflammation in murine models of Alzheimer’s disease [18]. Dereplication using high performance liquid chromatography coupled with high resolution mass spectrometry (HPLC-HRMS) in natural product research has recently been introduced as a key step in the identification of secondary metabolites in target extracts. The HPLC-MS methods were also applied to discover structure-based compounds from natural products [19]. Thus, HPLC-qTOF-MS analysis and activity-guided purification were attempted in this study to find biologically active substances from 70% EtOH extract of *H. japonicus* against Parkinson’s disease. 

## 2. Results and Discussion 

The chromatographic procedures (silica gel, Sephadex LH-20, RP-18 and HPLC) of the 70% EtOH extract of *H. japonicus* using the bioactive-guided method afforded 14 compounds, including one new (**1**) along with 13 known (**2**–**14**) as active principles (Figure 1). 

Compound **1** was obtained as a brown gum with ${\left[\alpha \right]}_{D}^{20}$ − 35.0 (*c* 0.1, MeOH). HRESIMS ion peak at *m/z* 393.1887 [M + Na]^+^ (calcd for C_19_H_30_O_7_Na, 393.1884) suggested a molecular formula of C_19_H_30_O_7_ with five double-bond equivalents (Appendix A). IR spectra of compound **1** showed absorptions of hydroxyl (3396 cm^−1^), *α*,*β*-unsaturated ketone (1648 cm^−1^), and olefin (1416 cm^−1^) functionalities. The ^1^H NMR spectrum of compound **1** (Table 1 and Appendix A) showed four methyl singlets (*δ*_H_ 1.20, 1.20, 1.29, and 2.28) and two olefinic protons at *δ*_H_ 5.91 (1H, br s) and *δ*_H_ 5.97 (1H, t, *J* = 7.5 Hz). The ^13^C NMR spectrum (Table 1 and Appendix A) revealed the presence of four olefinic carbons at *δ*_C_ 129.2, 129.5, 144.8, and 159.6 as well as an *α*,*β*-unsaturated ketone at *δ*_C_ 201.8. The coupling pattern of anomeric proton of compound **1** suggested that the glucopyranosyl (*δ*_H_ 4.39, *J* = 7.8 Hz) unit is in the *β* configuration, and the sugar unit is linked at C-3′ by a detailed analysis of the HSQC and HMBC spectra (Figure 2, Appendix A). These results indicated that compound **1** is similar to (*Z*)-4-[3′-(*β*-d-glucopyranosyloxy)butylidene]-3,5,5-trimethyl-2-cyclohexen-l-one known as megastigmane glucoside [20]. However, when compared of the known compound, downfield shifts at C-3′ (δ_C_ 77.9, +2.1 ppm), C-4′ (δ_C_ 22.0, +2.1 ppm) and C-1″ (δ_C_ 103.9, +1.4 ppm) of compound **1** were observed. These results show that the C-1″ configuration was determined by significant differences in chemical shifts near C-1″ (positions 3′, 4′ and 1″) [21]. Thus, compound **1** was determined as a new epimer of (*Z*)-4-[3′-(*β*-d-glucopyranosyloxy)butylidene]-3,5,5-trimethyl-2-cyclohexen-l-one.

The structures of 13 known compounds were determined to be (2*E*)-4-[benzoyl-oxy]-3-methyl-2-buten-1-yl-*β*-d-glucopyranosides (**2**) [22]; 1-(4-hydroxy-3-methoxy)-phenyl-2-[4-(1,2,3- trihydroxypropyl)-2-methoxy]-phenoxy-1, 3-propandiol (**4**) [23]; benzyl-*α*-l-arabinopyranosy l-(1″→6′)-*β*-d-glucopyranoside (**5**); phenylethyl-*α*-l-arabinopyranosyl-(1″→6′)-*β*-d-glucopyranoside (**6**) [24]; 2,3-dihydro-2-(4′-hydroxy-3′-methoxyphenyl)-3-(hydroxymethyl)-7-methoxy-5- benzofuranpropanol4′-*O*-*β*-glucopyranoside (**7**) [25]; (6R,9R)-3-oxo-*α*-ionol-*β*-d-glucopyranoside (**8**); (6R,9S)-3-oxo-*α*-ionol-*β*-d-glucopyranoside (**9**) [20]; and eugenyl-*β*-d-glucopyranoside (**10**) and also flavonoids; apigenin-7-*O*-glucoside **(3)** [26], luteolin-7-*O*-glucoside (**11**) [25], vitexin (**12**), apigenin (**13**) and luteolin (**14**) [27] by comparing their physico-chemical and spectroscopic data with those reported in the literature (Figure 1). 

In the HPLC-qTOF-MS data shown in Figure 3, two peaks numbered 3 and 11 were determined as the major components of *H. japonicus*. Peak 11 (15.94 min) showed a molecular ion peak at *m/z* 447.1017 [M − H]^−^ and 285.0408 [M – Glc-H]^−^, and was determined as luteoin-7-*O*-glucoside with a molecular weight 448. The peak 3 (17.46 min) with a molecular ion at *m/z* 431.1030 [M – H]^−^ was identified as luteoin-7-*O*-glucoside and these two compounds commonly showed loss of 162 mass units from the flavonoid skeleton [26]. 

Peak 16 with ions observed at *m/z* 609.1506 [M − H]^−^, 447.0973 [M − H-Glc]^−^ and 285.0291 [M − H-2Glc]^−^ indicated that there was the loss of two glucose moieties. Peak 16 was predicted to be a flavonoid diglycoside similar to peak 11. Peaks 18, 19, and 20 showed the same base ion peak at *m/z* 361.2007 [M − H]^−^, which has a molecular weight of 362. The predominant loss of 69 amu (-C_5_H_9_, prenyl group) as the MS fragment ion was estimated to be from the hop bitter acids, a constituent of the Humulus genus [28,29]. Thus, in comparison with the MS fragment ions and retention times in the literature [28], peaks 18, 19 and 20 were predicted to be humulone, adhumulone and adprehumulone. Peak 22 showed the presence of *m/z* 431.2011 [M − H + H_2_O]^−^_,_ indicating a molecular weight of 414. Peak 22 showed the loss of 69 (-C_5_H_9_) and 118 (-C_9_H_10_) amu as the prenyl group moiety, which is an isoprenyl group in bitter acids. The molecular weight, fragmentation ion and retention time of peak 22 suggested that it is lupulone. The authentic standards of all isolated compounds (**1**–**14**) were analyzed as reference materials using the HPLC-MS method, and their identities were confirmed based on a comparison of the retention times and the MS spectra. Dereplication of the active constituents from the active fraction of *H. japonicus* (HJF) using HPLC-qTOF-MS and NMR suggested the MS data including retention time, molecular formula, and fragment ion for the isolated compounds (**1**–**14**) and the predicted compounds (**15**–**23**) (Table 2). 

The contents of luteolin-7-*O*-glucoside and apigenin-7-*O*-glucoside in the extract and fraction (HJF) of *H. japonicus* were analyzed by the HPLC method (Appendix A). Each standard was obtained as follows: luteolin-7-*O*-glucoside (*y* = 2291.5884*x* + 85.4579, r^2^ = 0.9999, **11**), apigenin-7-*O*-glucoside (*y* = 1804.6794*x* − 0.7846, r^2^ = 1, **3**), eugenyl-*β*-d-glucopyranoside (*y* = 3082.641*x* + 2.5291, r^2^ = 0.999, **10**), vitexin (*y* = 1498.667*x* + 14.1166, r^2^ = 0.997, **12**), luteolin (*y* = 24,048.89*x −* 52.7167, r^2^ = 0.999, **13**) and apigenin (*y* = 1616.224*x* + 15.5083, r^2^ = 0.999, **14**). The contents of luteolin-7-*O*-glucoside and apigenin-7-*O*-glucoside were 13.4 μg/mg and 6.5 μg/mg in the 70% ethanol extract of *H. japonicus*, respectively. After Diaion^®^ HP-20 column chromatography (Mitsubishi Chemical Corporation, Tokyo, Japan) of the 70% ethanol extract, the contents of luteolin-7-*O*-glucoside and apigenin-7-*O*-glucoside in HJF increased to 168.4 and 62.8 μg/mg, respectively. The major compounds of HJF were luteolin-7-*O*-glucoside and apigenin-7-*O*-glucoside, and there was approximately 12.57- and 9.68-folds increase in the contents of these flavonoids compared to those of the 70% EtOH extract. Among 14 compounds, the contents of minor compounds eugenyl-β-d-glucopyranoside, vitexin, luteolin and apigenin in 70% ethanol extract were 0.40, 1.90, 0.24, and 0.29 μg/mg. In HJF, they were increased to 2.70, 10.4, 0.50, and 0.40 μg/mg, respectively. At the concentration measured, other compounds were not detected due to the limit of peak area absorption. 

We tested the MAO B inhibitory activities of 14 isolated compounds (20 μM) isolated from *H. japonicus*. Safinamide, a reversible MAO B inhibitor, was used as a positive control in the experiment [30]. DMSO was used as a control group to exclude the influence of the solvent. As shown in Figure 4, isolated compounds 12, 13, and 14 namely vitexin, apigenin and luteolin at 20 µM exhibited the greatest inhibition of MAO B, with enzyme inhibitory activities ranging from 58% to 79%, while compounds **3** and **11** showed weak inhibitory activities compared to the control group. 

To determine the effect of HJF on Parkinson’s disease, we used a mouse model with a unilateral 6-OHDA lesion. The unilateral dopaminergic neuronal death induced by 6-OHDA lesions in mice caused limb-use asymmetry in the cylinder [31]. This limb-use behavior was significantly reduced by unilateral 6-OHDA lesions in both vehicle-treated mice and HJF-treated mice (Figure 5A). However, the HJF-treated group showed a much greater use of the contralateral limb compared to the vehicle group (Figure 5A). 21 days after surgery, d-AMPH-induced rotational behavior was measured in the vehicle group and the HJF group. Unilaterally-caused dopaminergic cell death in the midbrain resulted in asymmetric rotational behavior in the mice. Figure 5 shows the rotational behavior of mice by following an injection of 5 mg/kg d-AMPH. 6-OHDA-lesioned mice displayed robust ipsilateral rotation in response to d-AMPH, while HJF-treated mice significantly decreased ipsilateral rotations (Figure 5B,C). These results indicate that oral administration of 300 mg/kg HJF leads to improved motor asymmetry and motor impairment in unilateral 6-OHDA-lesioned mice. 

As oxidative damage in the brain provokes mitochondrial dysfunction and damage, oxidative stress is considered key to the pathogenesis of Parkinson’s disease (PD). The 6-OHDA neurotoxin leads to the formation of reactive oxygen species (ROS) and causes cell death in neurons of an experimental animal model of PD. A previous study of 70% EtOH extract of *H. japonicus* demonstrated neuroprotective activity in 6-hydroxydopamine (6-OHDA) animal model, and alleviated dopaminergic cell death and fiber loss caused by 6-OHDA [17]. However, identification and characterization of active compounds against PD from the 70% EtOH extract of *H. japonicus* have not been reported. MAO-B inhibitory activity-guided fractionations from *H. japonicus* lead to the isolation of one new and 13 known compounds.

Compounds **2** and **10** were reported to have ACE (angiotensin I converting enzyme) inhibition activity and may decrease oxidative stress and inflammation [22]. Megastigmane glucoside, compound **8**, showed a cell protective effect on benzo[a]pyrene-induced cytotoxicity [32]. Flavonoids, luteolin-7-*O*-glucoside (LG, **11**), apigenin-7-*O*-glucoside (AG, **3**), vitexin, luteolin and apigenin with MAO-B inhibitory activity were isolated from HJF. LG and AG were major components of HJF, and there was a 12.57- and 9.68-fold increase in the HJF compared to those in 70% EtOH extracts of *H. japonicus*. Luteolin and apigenin have been reported to have microglia-mediated inflammatory effects by regulating the induction of CD40 in response with IFN-γ in N9 and murine-derived primary microglial cells [33]. Luteolin also reduced the pathology of A*β*-amyloid deposit due to traumatic brain injury (TBI) in animal models. In addition, luteolin also significantly inhibited GSK activation, tau phosphorylation and microglial-induced release of inflammatory cytokines [34]. The metabolism of flavonoids was studied in previously cultured intestinal Caco-2 cells and rat intestines. Flavonoid glycosides generally undergo deglucosylation by the lactase phlorizin hydrolase (LPH) and gut enzymes or bacteria localized to the intestinal barrier [35]. LG hydrolyzes to luteolin and produces aglycone through intestinal enzymes or bacteria. Metabolized luteolin from LG was absorbed in the digestive tract without any conjugated forms including glucuronides. The bioavailability of luteolin-7-*O*-glucoside was approximately 10%, and biotransformed luteolin was detected. In a pharmacokinetics study with rats, the ratio of biotransformation into luteolin was approximately 48.78% [36]. AG is also hydrolyzed enzymatically to apigenin by *β*-glucosidases in human small intestine [37]. The bioavailability of AG in a germ-free rat model suggested that the major metabolite of AG is apigenin [38]. Therefore, the metabolic pathway of AG may have a pattern similar to that of LG. Humulones and lupulone which were predicted by HPLC-qTOF-MS have been reported antioxidant effects in the literature [39] and are expected to have synergistic effects in *H. japonicus*. 

Previously, 70% EtOH extract of *H. japonicus* showed neuroprotective effects on dopaminergic neurons in 6-OHDA-lesioned mice [17]. Oral administration of 70% EtOH extract of *H. japonicus* at 300 mg/kg showed decreased d-AMPH-induced ipsilateral rotations, but these decreases were not significant. In this study, a strong motor improvement effect was observed in the 6-OHDA-lesioned PD model when the same dose of HJF was administered. The right forelimb damaged by unilateral 6-OHDA injection was effectively improved (Figure 5A). Moreover, rotations under the presynaptically active d-AMPH were markedly suppressed by the administration of 300 mg/kg HJF (Figure 5B,C). Although dopaminergic neuron death in the midbrain was not investigated in this study, rotation intensity was correlated with the degree of nigrostriatal denervation [40]. Thus, dereplication of the active constituents from HJF using HPLC-qTOF-MS and NMR resulted in purified compounds (**1**–**14**) and predicted compounds (**15**–**23**). In particular, the HJF showed a significant increase in contents of AG and LG compared to the 70% EtOH extract of *H. japonicus*. These results suggest that flavonoids may have a neuroprotective effect in various neurodegenerative diseases including Parkinson’s disease (PD). 

## 3. Materials and Methods

### 3.1. General Information

Biologically active fraction of *H. japonicus* against Parkinson’s disease (HJF) was analyzed by an HPLC-qTOF-MS system consisting of a 1260 HPLC system and a 6530 qTOF-MS system (Agilent Technol., Santa Clara, CA, USA). Semi-preparative HPLC was performed using a Gilson HPLC system with a 321 pump and a UV/VIS-155 detector. An RS Tech Optima Pak C_18_ column (10 × 250 mm, 10 μm particle size, Korea) was used as the HPLC column. NMR spectra for 1D (^1^H and ^13^C NMR) and 2D (NOESY, HSQC and HMBC) were collected using a Bruker 400 MHz, 500 MHz and a JEOL JNM-ECA 600 MHz NMR spectrometer. ZEOprep 60 Silica gel (40−63 μm particle size), Cosmosil 75C_18_–prep, Diaion^TM^ Ion exchange resin for HP-20, and GE Healthcare Sephadex^TM^ LH-20 (18–111 μm) were used for column chromatography. Thin layer chromatography was performed using silica gel 60 F_254_ and RP-18 F_254_ plates. All solvents for extraction and isolation were of analytical grade. 

### 3.2. Plant Material 

The aerial part of *H. japonicus* was collected in Gangwon province in the Republic of Korea in 2014 and was botanically identified by Professor W.K. Oh. The voucher specimen (SNU2014-08) was deposited at the College of Pharmacy, Seoul National University, Seoul, Korea.

### 3.3. MAO-B Inhibitory Activity-guided Fractionation and Isolation 

The aerial part of *H. Japonicus* (1 kg) was extracted with 70% EtOH (6 L × 3 times) at 60 °C by ultra-sonication. The combined extracts were concentrated to dryness via gentle evaporation, and the dry residue (105 g) was dissolved in 30% EtOH. The crude extract solution was applied to a Diaion^®^ HP-20 column with H_2_O/EtOH gradients to give three fractions (30% EtOH, 90% EtOH, acetone, 3 L, respectively). The 90% EtOH fraction (HJF, 10.2 g) with the strongest MAO inhibitory activity was subjected to a normal-phase (NP) silica gel column with CH_2_Cl_2_/MeOH gradients (from 10:1 to 1:10) to afford six subfractions (HJA 1–6). The subfraction HJA 2 was purified on NP silica gel with CH_2_Cl_2_/MeOH (from 3:1 to 1:1) to yield compounds **3** (125 mg), **11** (50 mg), and **12** (20 mg). The subfraction HJA 3 was applied to a Sephadex^TM^ LH-20 column with 80% MeOH to produce five fractions (HJA 3.1–3.5). The fraction HJA 3.1 was applied to RP-C_18_ silica gel with MeCN/H_2_O gradients (from 1:6–1:1) to give 6 subfractions (HJA 3.1.1–3.1.6). Further purification of HJA 3.1.2 and HJA 3.1.4 using a Gilson HPLC system [RS Tech Optima Pak RP-C_18_ column (10 ID × 250 mm, 10 µm particle size); MeOH/H_2_O (14/86) containing 0.1% formic acid; flow rate 2 mL/min; 205 and 254 nm UV detection] yielded compounds **4** (3.5 mg) and **5** (6.5 mg), respectively. The subfraction 3.1.6 from the Gilson HPLC system with MeOH/H_2_O (20/80) containing 0.1% formic acid (flow rate 2 mL/min; 205 nm and 254 nm UV detection) afforded compounds **6** (8 mg) and **7** (8.5 mg). Subfraction HJA 4 was loaded onto a Sephadex^TM^ LH-20 column with 70% MeOH to produce nine subfractions (HJA 4.1–4.9). Compounds **1**, **8**, and **9** (5.0 mg, 4.0 mg, and 6.0 mg, respectively) were obtained from the fraction HJA 4.6 with MeCN/H_2_O (21/79) isocratic condition on a Gilson HPLC system. HJA 4.9 was purified by a similar HPLC system with MeCN/H_2_O gradients (20/80 → 25/75) to yield compounds **2** and **10** (4.0 mg and 2.8 mg, respectively). The subfraction HJA1 was purified using a Sephadex^TM^ LH-20 column with 100% MeOH to obtain compounds **13** (10 mg) and **14** (12 mg). 

Compound **1**: Brown gum, ${\left[\alpha \right]}_{D}^{20}$ −35.0° (*c*= 0.1, MeOH). IR (KBr) *v*_max_: 3396, 2948, 2913, 2837, 1648, 1416, 1112, 1080 and 1026 cm^−1^. HRESIMS *m/z* 393.1887 [M + Na]^+^ (calcd for C_19_H_30_NaO_7_: 393.1884) and 415.1963 [M + HCOO]^−^ (calcd for C_20_H_31_O_9_: 415.1974). ^1^H and ^13^C NMR data (CD_3_OD, 500 and 125 MHz), see Table 1.

Compound **2**: ^1^H NMR (CD_3_OD, 600 MHz): δ 8.02 (2H, d, 7.8 Hz, H-2, 6), 7.59 (1H, t, 7.4 Hz, H-4), 7.47 (2H, t, 7.7 Hz, H-3, 5), 5.82 (1H, t, 6.7 Hz, H-2′), 4.91 (2H, d, 6.9, H-1′), 4.31 (1H, d, 12.7 Hz, H-4′a), 4.29 (1H, d, 7.8 Hz, H-1″), 4.12 (1H, d, 12.7 Hz, H-4′b), 3.85 (1H, dd, 11.8, 1.9 Hz, H-6″a), 3.67 (1H, dd, 11.8, 5.7 Hz, H-6″b), 1.84 (3H, s, H-5′); ^13^C NMR (CD_3_OD, 150 MHz): δ 139.5 (C-3′), 134.2 (C-4), 131.6 (C-1), 130.5 (C-2, 6), 129.6 (C-3, 5), 122.1 (C-2′), 103.2 (C-1″), 78.1 (C-3″), 78.0 (C-5″), 75.1 (C-2″), 71.7 (C-4″), 74.5 (C-4′), 62.8 (C-6″), 62.4 (C-1′), 14.3 (C-5′).

Compound **3**: ^1^H NMR (DMSO-*d_6_*, 500 MHz): δ 7.96 (2H, d, 9.6 Hz, H-2′, 6′), 6.98 (2H, d, 9.6 Hz, H-3′, 5′), 6.86 (1H, s, H-3), 6.83 (1H, d, 2.2 Hz, H-6), 6.45 (1H, d, 2.2 Hz, H-8), 5.06 (1H, d, 7.3 Hz, H-1″); ^13^C NMR (DMSO-*d_6_*, 75 MHz): δ 162.9 (C-2), 103.1 (C-3), 181.9 (C-4), 161.1 (C-5), 99.5 (C-6), 165.2 (C-7), 94.8 (C-8), 156.9 (C-9), 105.3 (C-10), 121.0 (C-1′), 128.6 (C-2′), 116.0 (C-3′), 161.3 (C-4′), 116.0 (C-5′), 128.6 (C-6′), 99.9 (C-1″), 73.1 (C-2″), 76.4 (C-3″), 69.6 (C-4″), 77.2 (C-5″), 60.6 (C-6″).

Compound **4**: ^1^H NMR (CD_3_OD, 600 MHz): δ 7.07 (1H, d, 1.6 Hz, H-2′), 7.03 (2H, m, H-2, 5′), 6.88 (1H, d, 8.2 and 2.0 Hz, H-6), 6.86 (1H, dd, 8.4 and 1.3 Hz, H-6′), 6.72 (1H, d, 8.2 Hz, H-5), 4.89 (1H, d, 6.0 Hz, H-7), 4.58 (1H, d, 6.0 Hz, H-7′), 4.29 (1H, m, H-8), 3.88 (3H, s, OMe), 3.82 (3H, s, OMe), 3.73 (1H, dd, 11.5 and 4.1 Hz, H-9a), 3.66 (1H, m, H-8′), 3.51 (1H, dd, 11.4 and 4.2 Hz, H-9′a), 3.48 (1H, dd, 11.5 and 3.5 Hz, H-9b), 3.37 (1H, dd, 11.4 and 6.3 Hz, H-9′b); ^13^C NMR (CD_3_OD, 150 MHz): δ 133.8 (C-1), 111.7 (C-2), 151.5 (C-3), 148.9 (C-4), 118.7 (C-5), 120.7 (C-6), 74.0 (C-7), 87.2 (C-8), 61.9 (C-9), 137.9 (C-1′), 112.3 (C-2′), 148.8 (C-3′), 147.2 (C-4′), 115.8 (C-5′), 120.6 (C-6′), 75.1 (C-7′), 77.4 (C-8′), 64.2 (C-9′), 56.4 and 56.5 (OMe).

Compound **5**: ^1^H NMR (CD_3_OD, 500 MHz): δ 7.44 (2H, d, 7.3 Hz, H-2, 6), 7.35 (2H, d, 7.2 Hz, H-3, 5), 7.28 (1H, m, H-4), 4.93 (1H, d, 11.8 Hz, H-7a), 4.68 (1H, d, 11.8 Hz, H-7b), 4.37 (2H, d, 7.6 Hz, H-1′, 1″); ^13^C NMR (CD_3_OD, 125 MHz): δ 139.1 (C-1), 129.3 (C-2, 6), 129.2 (C-3, 5), 128.7 (C-4), 71.6 (C-7), 103.3 (C-1′), 77.1 (C-2′), 78.0 (C-3′), 71.9 (C-4′), 77.7 (C-5′), 71.2 (C-6′), 105.6 (C-1″), 74.9 (C-2″), 75.1 (C-3″), 69.8 (C-4″), 66.9 (C-5″).

Compound **6**: ^1^H NMR (CD_3_OD, 400 MHz): δ 7.32 (2H, s, H-2, 6), 7.31 (2H, s, H-3, 5), 7.22 (1H, m, H-4), 4.37 (1H, d, 7.6 Hz, H-1″), 4.36 (1H, d, 7.6 Hz, H-1′), 4.11 (1H, m, H-8a), 3.80 (1H, m, H-8b), 2.99 (2H, t, 7.3 Hz, H-7); ^13^C NMR (CD_3_OD, 100 MHz): δ 140.1 (C-1), 130.0 (C-2, 6), 129.3 (C-3, 5), 127.2 (C-4), 37.2 (C-7), 71.5 (C-8), 105.5 (C-1′), 74.9 (C-2′), 77.9 (C-3′), 71.5 (C-4′), 77.0 (C-5′), 69.8 (C-6′), 104.4 (C-1″), 75.0 (C-2″), 77.7 (C-3″), 71.2 (C-4″), 66.9 (C-5″).

Compound **7**: ^1^H NMR (CD_3_OD, 600 MHz): δ 7.15 (1H, d, 8.1 Hz, H-5′), 7.03 (1H, br s, H-2′), 6.94 (1H, d, 8.1 Hz, H-6′), 6.74 (1H, s, H-2), 6.72 (1H, s, H-6), 5.56 (1H, d, 5.7 Hz, H-7′), 3.86 (3H, s, OMe), 3.83 (3H, s, OMe), 3.77-3.39 (9H, m), 3.57 (2H, t, 6.1 Hz, H-9), 2.63 (2H, t, 7.4 Hz, H-7), 1.81 (2H, m, H-8); ^13^C NMR (CD_3_OD, 150 MHz): δ 129.5 (C-1), 117.9 (C-2), 145.2 (C-3), 147.4 (C-4), 137.0 (C-5), 114.0 (C-6), 32.9 (C-7), 35.8 (C-8), 62.4 (C-9), 138.3 (C-1′), 111.0 (C-2′), 147.5 (C-3′), 150.9 (C-4′), 117.9 (C-5′), 119.3 (C-6′), 88.4 (C-7′), 55.6 (C-8′), 65.0 (C-9′), 102.7 (C-1″), 74.9 (C-2″), 78.2 (C-3″), 71.3 (C-4″), 77.8 (C-5″), 62.5 (C-6″).

Compound **8**: ^1^H NMR (CD_3_OD, 500 MHz): δ 5.88 (1H, s, H-4), 5.80 (1H, dd, 15.4 and 6.4 Hz, H-8), 5.67 (1H, dd, 15.4 and 8.6 Hz, H-7), 4.41 (1H, m, H-9), 4.36 (1H, d, 6.5 Hz, H-1′), 2.67 (1H, d, 9.3 Hz, H-6), 2.45 (1H, d, 16.7 Hz, H-2a), 2.06 (1H, d, 16.7 Hz, H-2), 1.30 (1H, d, 6.3 Hz, H-10), 1.94 (3H, s, H-13), 1.03 (3H, s, H-11), 1.01 (3H, s, H-12); ^13^C NMR (CD_3_OD, 125 MHz): δ 37.1 (C-1), 202.0 (C-3), 126.1 (C-4), 165.9 (C-5), 56.8 (C-6), 128.8 (C-7), 138.2 (C-8), 78.0 (C-9), 21.0 (C-10), 28.1 (C-11), 27.6 (C-12), 23.8 (C-13), 102.5 (C-1′), 75.2 (C-2′), 78.1 (C-3′), 71.5 (C-4′), 77.0 (C-5′), 62.7 (C-6′).

Compound **9**: ^1^H NMR (CD_3_OD, 400 MHz): δ5.88 (1H, br s, H-4), 5.77 (1H, dd, 15.4 and 9.3 Hz, H-7), 5.61 (1H, dd, 15.4 and 7.4 Hz, H-8), 4.56 (1H, m, H-9), 4.29 (1H, d, 7.7 Hz, H-1′), 2.70 (1H, d, 9.3 Hz, H-6), 2.48 (1H, d, 16.8 Hz, H-2a), 2.07 (1H, d, 16.8 Hz, H-2b), 2.04 (1H, d, 1.1 Hz, H-13), 2.04 (3H, d, 1.1 Hz, H-13), 1.02 (3H, s, H-11), 0.98 (3H, s, H-12); ^13^C NMR (CD_3_OD, 100 MHz): δ 38.0 (C-1), 202.8 (C-3), 127.0 (C-4), 166.4 (C-5), 57.7 (C-6), 132.0 (C-7), 137.9 (C-8), 75.6 (C-9), 23.0 (C-10), 28.2 (C-11), 28.8 (C-12), 24.7 (C-13), 102.0 (C-1′), 75.8 (C-2′), 79.2 (C-3′), 72.5 (C-4′), 79.0 (C-5′), 63.7 (C-6′).

Compound **10**: ^1^H NMR (DMSO-*d_6_*, 500 MHz): δ 6.99 (1H, d, 8.2 Hz, H-5), 6.79 (1H, d, 1.9 Hz, H-2), 6.66 (1H, dd, 8.2, 1.9 Hz, H-6), 5.94 (1H, m, H-8), 5.06 (1H, dd, 17.6, 1.6 Hz, H-9b), 5.02 (1H, dd, 9.6, 1.6 Hz, H-9a), 4.84 (1H, d, 7.3 Hz, H-1′), 3.24 (1H, m, H-7b), 3.14 (1H, m, H-7a); ^13^C NMR (DMSO-*d_6_*, 125 MHz): δ 148.85 (C-3), 144.84 (C-4), 137.91 (C-1), 133.41 (C-8), 120.27 (C-6), 115.52 (C-2), 115.47 (C-9), 112.86 (C-5), 100.20 (C-1′), 76.97 (C-3′), 76.84 (C-5′), 73.21 (C-2′), 69.66 (C-4′), 60.65 (C-6′), 55.599 (C-10), 40.41 (C-7)

Compound **11**: ^1^H NMR (DMSO-*d_6_*, 500 MHz): δ 7.45 (1H, dd, 8.4, 2.3 Hz, H-2′), 7.41 (1H, d, 2.2 Hz, H-6′), 6.90 (1H, d, 8.4 Hz, H-3′), 6.78 (1H, d, 2.1 Hz, H-6), 6.76 (1H, s, H-3), 6.44 (1H, d, 2.1 Hz, H-8), 5.08 (1H, d, 7.5 Hz, H-1″); ^13^C NMR (DMSO-*d_6_*, 125 MHz): δ 182.3 (C-4), 164.9 (C-2), 163.4 (C-7), 161.6 (C-5), 157.4 (C-9), 150.4 (C-4′), 146.2 (C-3′), 121.8 (C-1′), 119.6 (C-6′), 116.4 (C-5′), 114.0 (C-2′), 105.8 (C-10), 103.6 (C-3), 100.3 (C-1″), 99.9 (C-6), 95.1 (C-8), 77.6 (C-5″), 76.8 (C-3″), 73.5 (C-2″), 69.9 (C-4″), 61.0 (C-6″)

Compound **12**: ^1^H NMR (DMSO-*d_6_*, 500 MHz): δ 8.13 (2H, d, 8.3 Hz, H-2′, 6′), 7.02 (2H, d, 8.3 Hz, H-3′, 5′), 6.73 (1H, s, H-3), 6.39 (1H, s, H-6), 4.91 (1H, d, 9.8 Hz, H-1″); ^13^C NMR (DMSO-*d_6_*, 125 MHz): δ 183.1 (C-4), 165.0 (C-2), 163.7 (C-7), 162.2 (C-4′), 161.7 (C-5), 157.1 (C-9), 129.7 (C-2′, 6′), 122.7 (C-1′), 116.6 (C-3′, 5′), 105.4 (C-8), 105.0 (C-10), 103.2 (C-3), 99.0 (C-6), 82.5 (C-5″), 79.9 (C-3″), 74.4 (C-1″), 72.0 (C-2″), 71.6 (C-4″), 62.4 (C-6″)

Compound **13**: ^1^H NMR (DMSO-*d_6_*, 500 MHz): δ 7.93 (2H, d, 8.8 Hz, H-2′, 6′), 6.93 (2H, d, 8.8 Hz, H-3′, 5′), 6.77 (1H, s, H-3), 6.46 (1H, s, H-6), 6.17 (1H, s, H-8); ^13^C NMR (DMSO-*d_6_*, 125 MHz): δ 164.1 (C-2), 103.2 (C-3), 182.1 (C-4), 161.7 (C-5), 99.4 (C-6), 165.1 (C-7), 94.5 (C-8), 157.8 (C-9), 104.0 (C-10), 121.6 (C-1′), 128.9 (C-2′, 6′), 116.4 (C-3′, 5′), 161.9 (C-4′).

Compound **14**: ^1^H NMR (DMSO-*d_6_*, 500 MHz): δ 7.40 (1H, d, 2.2 Hz, H-2′), 7.42 (1H, dd, 8.3, 2.3 Hz, H-6′), 6.89 (1H, d, 8.3 Hz, H-5′), 6.68 (1H, s, H-3), 6.45 (1H, d, 2.0 Hz, H-6), 6.19 (1H, d, 2.0 Hz, H-8); ^13^C NMR (DMSO-*d_6_*, 125 MHz): δ 164.3 (C-2), 103.3 (C-3), 182.1 (C-4), 161.9 (C-5), 99.3 (C-6), 164.9 (C-7), 94.3 (C-8), 157.7 (C-9), 104.1 (C-10), 119.4 (C-1′), 113.8 (C-2′), 146.2 (C-3′), 150.2 (C-4′), 116.5 (C-5′), 121.9 (C-6′).

### 3.4. HPLC-qTOF-MS/MS Analysis

Chromatographic analysis was performed on an Agilent 1260 Series LC system (Agilent Technologies Co. Ltd., Waldbronn, Germany) equipped with a binary pump, degasser, an automatic sampler and a thermostatically controlled column oven. The mobile phase consisted of MeCN with 0.1% formic acid (A) and water with 0.1% formic acid (B) using gradient elution from 0–3 min (10% A), 3–43 min (10→90% A), 43–55 min (100% A) and 55–60 min (10% A). The flow rate was 0.3 mL/min and analyzed by injecting 10 μL of a 0.1 mg/mL sample. Detection was performed using a 6530 qTOF mass spectrometer (Agilent Technologies Co. Ltd.) equipped with an ESI interface. The conditions of the ESI source were as follows: gas temperature, 350 °C; drying gas, 10 L/min; fragmentor, 180 V; nebulizer, 30 psi; skimmer, 60 V; sheath gas temperature, 350 °C; sheath gas flow, 12 L/min; OCT IRF V_pp_, 750 V; V_cap_, 4000 V; and collision energy, 10 V for MS/MS analysis. The internal references for mass showed ions at *m/z* 119.0363 and 966.0007 in negative ion mode. Data acquisition and analysis were performed using Agilent MassHunter software (Agilent Technol., Santa Clara, CA, USA).

### 3.5. In vitro MAO-B Inhibition Assay

The reaction was initiated by incubating the test compound at the desired concentrations with kynuramine (Sigma Chemical Co., St Louis, MO, USA) in 0.1 M potassium phosphate buffer (pH 7.4) in a 96-well plate. After incubation at 37 °C for 10 min, 50 μL of MAO B enzyme (Sigma Chemical Co.) in 0.1 M potassium phosphate buffer was added to each reaction well. The reaction progress was terminated with the addition of 75 μL of 2 M sodium hydroxide after 20 min. The fluorescence intensity of 4-hydroxyquinoline was measured at an excitation wavelength of 310 nm and an emission wavelength of 400 nm using a SpectraMax GEMNI XPS microplate reader (Molecular Devices Corporation, Sunnyvale, CA, USA) [41]. The inhibition ratio is expressed as the activity percentage compared to the vehicle control. 

### 3.6. In vivo Animal Test

#### 3.6.1. Animals 

Nine-week-old male C57BL/6J mice were provided by the KRIBB and housed in a 22 ± 1 °C and humidity-controlled (50–60%) environment under specific pathogen-free conditions under 12-h light-dark cycles. The mice were randomly divided into two groups: vehicle control (0.5% carboxymethyl cellulose, *n* = 10) and HJF (300 mg/kg/day, *n* = 13). The vehicle or HJF was administrated daily by oral gavage for three days before 6-OHDA lesion and 21 days after 6-OHDA lesion. All animal experiments were approved by the Institutional Animal Care and Use Committee (IACUC) of KRIBB and were performed in accordance with the institutional guidelines of KRIBB. 

#### 3.6.2. 6-Hydroxydopamine (6-OHDA) Lesion

The procedure for 6-OHDA lesioning was previously described [31]. Mice were anesthetized with a mixture of ketamine hydrochloride (Yuhan corporation, Seoul, Korea) and xylazine hydrochloride (Bayer Korea) and mounted in a stereotactic frame (Stoelting Europe, Dublin, Ireland) equipped with a mouse adaptor. Mice were pretreated with desipramine hydrochloride [Sigma Aldrich, 25 mg/kg, intraperitoneal (i.p.)] 30 min before surgery to prevent noradrenergic neuronal damage, and unilaterally injected 6-OHDA in a volume of 3 μL (Sigma-Aldrich Co. LLC, 2 μg/μL diluted in saline containing 0.2% ascorbic acid) into the left dorsal striatum at the following coordinates: anteroposterior (AP), 1.2 mm; lateral, −1.8 mm; and dorsoventral, −3.6 mm. The mice were kept on a warming plate until they awoke from the anesthesia and were subsequently returned to their cages until they were used. To avoid dehydration, the mice received glucose (JW Pharmaceutical, Korea) in saline (10 mL/kg, s.c.) during the procedure. Additionally, in the evening of the first week after surgery, the food pellets were soaked in water and were placed in a shallow vessel on the floor of the cages. 

#### 3.6.3. Cylinder Test 

Individual mice were placed into a transparent acrylic cylinder (diameter, 20 cm) and recorded for 5 min. Tests were conducted before and 7 days after 6-OHDA injection. The number of times that the right and left forelimbs contacted the wall was calculated by the blind observer. Use of the impaired (right) forelimb was expressed as a percentage of the total number of supporting wall contacts [31]. 

#### 3.6.4. D-AMPH-induced Rotation Test 

Dextro-amphetamine (d-AMPH, USP, Rockville, MD, USA, 5 mg/kg, i.p.)-induced rotation was measured 21 days after 6-OHDA injection. The ipsilateral turning behavior induced by d-AMPH administration was recorded for 60 min. The number of ipsilateral rotations was analyzed by a SMART video tracking system (Panlab, Spain). 

### 3.7. Statistical Analysis

The results are expressed as the means ± SD through three independent experiments. Statistical analysis was performed using Microsoft Excel 2010. Differences between groups were compared using one-way analysis of variance followed by a two-tailed Student’s t-test for unpaired samples, assuming equal variance (* *p* < 0.05, ** *p* < 0.01, *** *p* < 0.001, compared to the control). 

## 4. Conclusions 

The 70% EtOH extract of *H. japonicus* previously reported neuroprotective effects on dopaminergic neurons in 6-OHDA-lesioned mice. In this study, we observed a stronger motor improvement effect in the 6-OHDA-lesioned PD model when the same dose of the active enrichment fraction (HJF) of *H. japonicas* was used. The right forelimb damaged by unilateral 6-OHDA injection was effectively improved, and the rotations under the presynaptically active d-AMPH were markedly suppressed by the administration of 300 mg/kg HJF. Thus, intensive studies on the chemical profiling of HJF using HPLC-qTOF-MS and NMR resulted in isolation of fourteen compounds (**1**–**14**) including one new compound (**1**) and in prediction of nine compounds (**15**–**23**). Specifically, AG and LG contents were significantly increased in the HJF compared with those in the 70% EtOH extract of *H. japonicus*. These results suggest that AG and LG may have a neuroprotective effect in various neurodegenerative diseases including Parkinson’s disease (PD).

## Figures and Tables

**Figure 1 molecules-24-01435-f001:**
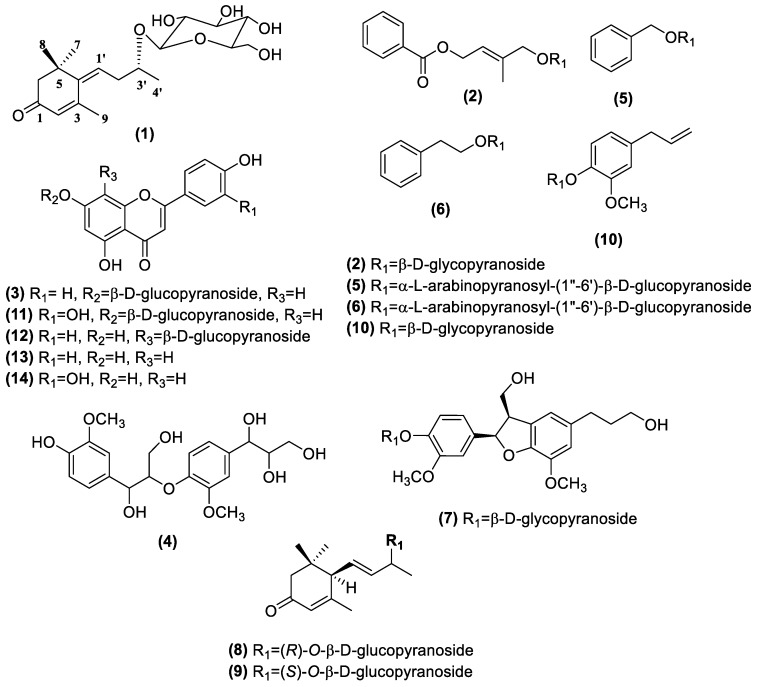
Chemical structures of compounds **1**–**14** isolated from the active fraction of *Humulus japonicus* (HJF).

**Figure 2 molecules-24-01435-f002:**
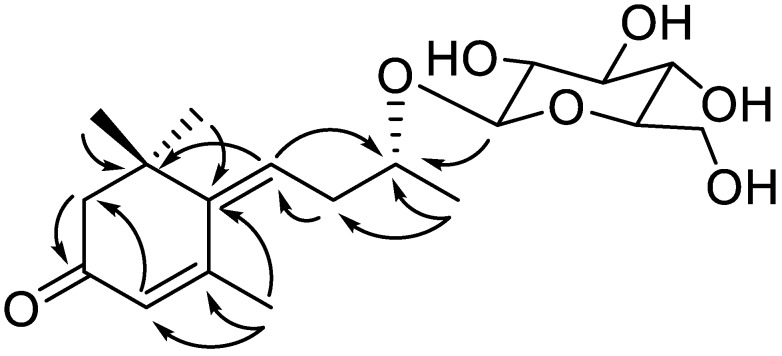
Key HMBC correlations (from H to C) for compound **1**.

**Figure 3 molecules-24-01435-f003:**
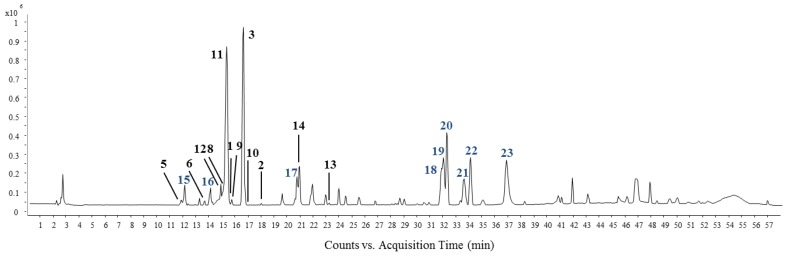
Total ion chromatogram of HJF by HPLC-ESI-qTOF-MS/MS in the negative ion mode. Peak assignments with isolated (**1**–**14**) and predicted (**15**–**23**) compounds.

**Figure 4 molecules-24-01435-f004:**
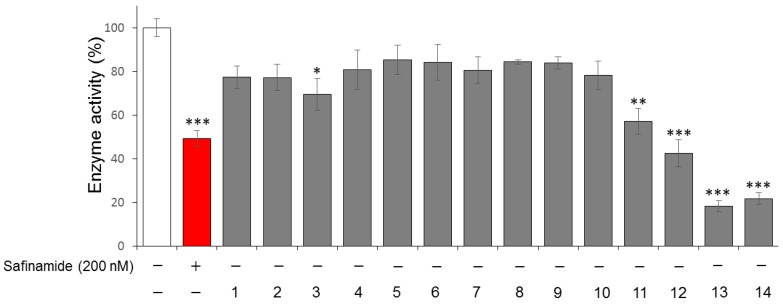
Inhibitory effect of 14 isolated compounds from *H. Japonicus* on MAO B based on fluorescent characteristics of 4-hydroxyquinoline, which is the product of the redox reaction catalyzed by MAO B. The inhibitory activity against MAO B was determined after MAO B was incubated with 14 compounds isolated from *H. Japonicus* at 20 µM. Values represent the relative activity of MAO B compared to the control group without addition of these inhibitory compounds as the mean ± SD (n = 3), * *p* < 0.05, ** *p* < 0.01 and *** *p* < 0.001.

**Figure 5 molecules-24-01435-f005:**
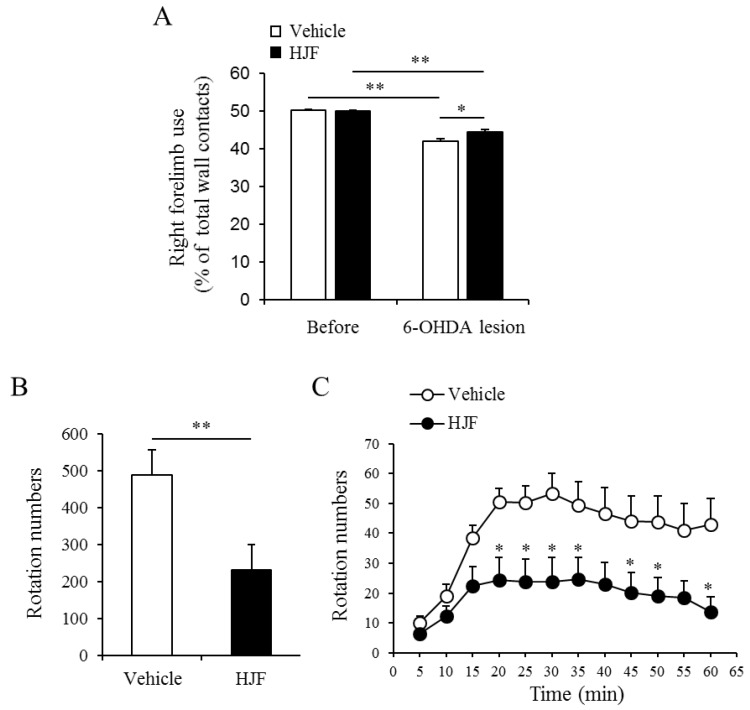
Oral administration of HJF improves forelimb-use asymmetry and d-AMPH-induced rotational asymmetry in a mouse model of PD. (**A**). Cylinder test: 6-OHDA-lesioned mice showed a significant decrease in the percentage of contralateral contacts of a forelimb at 7 days. HJF-treated group showed an increase in the percentage of contralateral contacts compared to that in the vehicle-treated group. * *p* < 0.05 and ** *p* < 0.01, one-way ANOVA followed by Tukey’s *post hoc* test. B and C. D-AMPH-induced rotation test. D-AMPH (5 mg/kg)-induced rotation was tested at 3 weeks after 6-OHDA lesion. The total rotation number was counted for 60 min after d-AMPH administration (**B**). Time course of the rotations (**C**). * *p* < 0.05 and ** *p* < 0.01, Student’s t-test.

**Table 1 molecules-24-01435-t001:** ^1^H and ^13^C NMR spectral data for compound **1** (*δ* in ppm, *J* in Hz).

Position	1 ^a^	
	*δ* _H_	*δ* _C_
1		201.8, C
2	5.91, br s	129.5, CH
3		159.6, C
4		144.8, C
5		42.0, C
6	2.31, d (6.0)	53.7, CH_2_
7	1.20, s	28.2, CH_3_
8	1.20, s	28.3, CH_3_
9	2.28, br s	25.1, CH_3_
1′	5.97, t (7.50)	129.2, CH
2′	2.55, m 2.64, m	38.0, CH_2_
3′	4.01, m	77.9, CH
4′	1.29, d (6.0)	22.0, CH_3_
1″	4.39, d (7.8)	103.9, CH
2″	3.18, t (8.0)	75.3, CH
3″	3.50–3.54, overlap	78.2, CH
4″	3.44, m	71.7, CH
5″	3.50–3.54, overlap	78.0, CH
6″	3.88, br d (11.8) 3.67, dd (11.8, 6.0)	62.8, CH_2_

***^a^*** Recorded in methanol*-d*4 at 500 MHz for proton and 125 MHz for carbon.

**Table 2 molecules-24-01435-t002:** Identification of isolated (**1**–**14**) and predicted (**15**–**23**) compounds from HJF by HPLC-qTOF-MS/MS analysis in negative ion mode.

Comp NO ^a^	t_R_ (min)	Molecular Formula	[M − H]^−^/[M + HCOO]^−^	Fragment Ion	Identification
1	15.352	C_19_H_30_O_7_	415.1975	339.1388, 271.1549, 223.0263	(*Z*)-6-[9-(β-d-glucopyranosyloxy)butylidene]-5,1,1-trimethyl-4-cyclohexen-3-one
2	17.885	C_18_H_24_O_8_	413.1446	343.2107, 299.1835	(2*E*)-4-[benzoyl-oxy]-3-methyl-2-buten-1-yl-*β*-d-glucopyranoside,
3	16.495	C_21_H_20_O_10_	431.1030	268.0363, 176.0077	Apigenin-7-*O*-β-d-glucopyranoside
5	11.876	C_18_H_26_O_10_	447.1689	315.0884, 269.0979, 191.0557	Benzyl-α-l-arabinopyranosyl-(1″→6′)-β-d-glucopyranoside
6	13.515	C_19_H_28_O_10_	461.1689	415.1607	Phenylethyl-α-l-arabinopyranosyl-(1″→6′)-β-d-glucopyranoside
8	14.955	C_19_H_30_O_7_	415.1964	284.0315, 223.0289, 130.9644	(*6R*,*9R*)-3-Oxo-*α*-ionol-*β*-d-glucopyranoside
9	15.601	C_19_H_30_O_7_	415.1989	369.1907, 223.0259, 119.0389	(*6R*,*9S*)-3-Oxo-*α*-ionol-*β*-d-glucopyranoside
10	16.931	C_16_H_22_O_7_	371.1354	163.0738	Eugenyl-β-d-glucopyranoside
11	15.203	C_21_H_20_O_11_	447.0990	377.1665, 284.0322, 151.0022	Luteolin-7-*O*-β-d-glucopyranoside
12	14.756	C_21_H_20_O_10_	431.0996	311.0547, 283.1125	Vitexin
13	23.100	C_15_H_10_O_5_	269.0463	227.1268, 183.1374	Apigenin
14	20.816	C_15_H_10_O_6_	285.0412	211.1314, 171.1022	Luteolin
15	11.975		431.1953	385.1836, 264.1225	Unknown
16	13.962	C_27_H_30_O_16_	609.1506	447.0973, 285.0291	Flavonoid diglucosides
17	20.617		327.2200	211.1321, 119.0367	Unknown
18	31.742	C_21_H_30_O_6_	361.2007	293.2150, 265.1484	Humulone
19	31.891	C_21_H_30_O_6_	361.2006	293.2149, 275.2023	Adhumulone
20	32.139	C_21_H_30_O_6_	361.2003	293.21516, 246.1214	Adprehunumolne
21	33.480		687.31692	623.2733, 555.2895	Unknown
22	33.977	C_26_H_38_O_4_	431.2019	363.2112, 295.2307	Lupulone
23	36.758		389.2071	321.2218, 293.1807	Unknown

^a^ compounds **4** and **7** were not detected by HPLC-qTOF-MS/MS analysis.

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
