# Peer review of "Dereplication of Components Coupled with HPLC-qTOF-MS in the Active Fraction of Humulus japonicus and It’s Protective Effects against Parkinson’s Disease Mouse Model"

_molecules, 2019, doi:10.3390/molecules24071435_

Reviewer 1 Report

The manuscript entitled "Dereplication of active constituents against Parkinson's disease from Humulus japonicus using High-Performance Liquid Chromatography(HPLC) with Quadruple-Time-of Flight Mass Spectroscopy (qTOF-MS) and NMR Techniques" by Won Keun Oh et al., is reported 14 compounds isolated from HJF, and investigated the possible effect of HJF in unilateral 6-OHDA-lesioned mice. However, the authors need to consider the following comments in their revised version.

1. The title should be revised to fit the main outputs of the study. 14 compounds were identified in this paper, the experiment of 6-OHDA mouse model shows that HJF has possible against PD effect, but it is performed offline using HJF, it can’t explain which compounds are the active components.

2. Page 5 Line 158-160. The author points out that the content of 3 and 11 compounds increases after 70% of the extracts pass through the column, but it does not explain the change and content of other compounds. It is suggested to measure and calculate the concentration of 14 compounds but not percent in HJF.

3. In the MAO-B inhibition experiment, the result showed the areas of compound 3, 11 in Fig 3 are relatively high, and compound 12,13,14 exhibited inhibition effect of MAO-B at 20 µM in Fig 4, but it is unknown that the concentration of compounds in the target tissue. It is suggested that the concentration of compounds in target tissues be provided to show whether the inhibitory concentration (20 µM) can be achieved?

Author Response

1. The title should be revised to fit the main outputs of the study. 14 compounds were identified in this paper, the experiment of 6-OHDA mouse model shows that HJF has possible against PD effect, but it is performed offline using HJF, it can’t explain which compounds are the active components.

As two reviewers 1 and 2’s comments, the title of this paper was revised like

Derelication of components coupled with HPLC-qTOF-MS in the active fraction of Humulus japonicus and its protective effects against Parkinson’s disease mouse model

2. Page 5 Line 158-160. The author points out that the content of 3 and 11 compounds increases after 70% of the extracts pass through the column, but it does not explain the change and content of other compounds. It is suggested to measure and calculate the concentration of 14 compounds but not percent in HJF.

→ As reviewer’s comment, we measured and calculated other compounds by quantitative analysis. The contents of four compounds (compounds 10, 12, 13, and 14) were newly described in Page 5 Line 156-170 of revised manuscript. However, at the concentration we measured, the remaining compounds had very low peak area and could not be measured due to the limit of calibration.

3. In the MAO-B inhibition experiment, the result showed the areas of compound 3, 11 in Fig 3 are relatively high, and compound 12,13,14 exhibited inhibition effect of MAO-B at 20 µM in Fig 4, but it is unknown that the concentration of compounds in the target tissue. It is suggested that the concentration of compounds in target tissues be provided to show whether the inhibitory concentration (20 µM) can be achieved?

→ As reviewer’s comment, authors et al., also think that the quality of our manuscript will be better if concentrations of the compounds and their MAO-B inhibitory effect of the compounds in target tissues should be analyzed. However, to get more accurate and informative insight for the compound effects on MAO inhibition in target tissue, the authors would think that the pharmacokinetic and pharmacodynamic investigations for some compounds will be considered concomitantly. However, authors have a limited time to prove this issue in the present study, and expect and promise that it can be achieved through the next round of independent experiment.

Reviewer 2 Report

In the manuscript, the authors investigated the potentially active components against Parkinson’s disease from H. japonicus fraction (HJF) using HPLC-QTOF/MS and NMR, and also revealed at least two compounds from HJF, apigenin and luteolin exhibited the strongest inhibitory effects on monoamine oxidase (MAO) B enzyme activity; Meanwhile, the authors also did some other activity test using some mice models. The work is interesting, however, it is not well presented in its current form. Thus, major revisions are suggested.

Some detailed comments are made as below:

1. The title of the manuscript should be revised to better reflect the major findings of the work, in fact, more chemical profiling work and less activity study were conducted in this work, and  the title claimed too much in its current form.

2. From Figure 3 and figure 4, it is obvious that there are at least 8 other compounds were not tested for activity, and dereplication of active constituents against Parkinson’s Disease from Humulus japonicas the authors claimed is not true and accurate at this point.

3. In figure 4, it seemed that compounds 13 and 14 showed better activities than that of others, why did the authors not use the subfraction HJA1 for further activity test, but use HJF?

4. The English expression throughout the text should be double checked and revised for accuracy and clarity. 

Author Response

1. The title of the manuscript should be revised to better reflect the major findings of the work, in fact, more chemical profiling work and less activity study were conducted in this work, and  the title claimed too much in its current form.

As two reviewers 1 and 2’s comments, the title of this paper was revised like

Derelication of components coupled with HPLC-qTOF-MS in the active fraction of Humulus japonicus and its protective effects against Parkinson’s disease mouse model

2. From Figure 3 and figure 4, it is obvious that there are at least 8 other compounds were not tested for activity, and dereplication of active constituents against Parkinson’s Disease from Humulus japonicas the authors claimed is not true and accurate at this point.

→ The 14 compounds were directly isolated, but remaining 8 compounds were predicted compounds using HPLC-qTOF-MS. As reviewer’s comments, we could not test them through experiments because we did not have these predicted 8 compounds by direct isolation. However, the correlation between compounds 3 and 11, which have moderate activity on MAO-B assay, and 13 and 14, which exhibits strong MAO-B activity, was described in Page 9 Line29-38 through direct activities on MAO-B assay. Compounds humulones and lupulones, which were predicted by HPLC-qTOF-MS, have been reported antioxidant effects in the other paper, and are expected to have synergistic effects. This discussion was newly commented in page 9 line 39-40 of revised manuscript.

3. In figure 4, it seemed that compounds 13 and 14 showed better activities than that of others, why did the authors not use the subfraction HJA1 for further activity test, but use HJF?

→ The authors are grateful for the reviewer’s comments like as “compounds 13 and 14 showed better activities than that of others, why did the authors not use the subfraction HJA1 for further activity test, but use HJF?”. Is this not tested with the HJA1 fraction containing more compounds 13 and 14? This is a very interesting and critical reviewer’s question. We also agree with the reviewer's questions. However, in order to answer this question, further studies on the present active fraction or HJA1 fraction should be conducted.

Especially, the cost of preparing the active fraction should be considered. Furthermore, there are many compounds in the present active fraction, and the synergistic effect by the combination of these compounds in the present active fraction should be verified.

As discussed in line 29-38 on page 9, humulone (18) and lupulone (22), which we predicted to exist in this active fraction through HPLC coupled with qTOP-MS method, had been suggested to have neuroprotective activity through other mechanisms. For example, in the new active fraction HJA1 in which compounds 13 and 14 are concentrated, the absence of compounds 18 and 22 will be effected the synergy of chemical constituents in present active fraction. We think that the reviewers' point of view is one of the subjects to be approached through further research in the future, and I am very grateful for this point of view.

4. The English expression throughout the text should be double checked and revised for accuracy and clarity. 

This paper already edited by American Journal Experts (AJE, www.aje.com) before submission. In addition, as reviewer’s comment, additional professional English professor checked again corrections about English problems in the revised manuscript. We expect our efforts to be evaluated with the reviewer’s point of view.

Round  2

Reviewer 1 Report

The authors have made some revised and promised to supplement the relevant data for comment 3. If the authors can provide some information about the concentration of compounds in target tissues to show whether the inhibitory concentration (20 µM) can be achieved, it is helpful for readers to understand the protective effects of compounds. I think the revised manuscript will meet with approval.

Reviewer 2 Report

Since the authors have revised the manuscript to address most if not all of the points raised by the reviewers, and the revised manuscript is now significantly improved. Thus, acceptance of the work is suggested.